# Patterns of Care and Data Quality in a National Registry of Black and White Patients with Merkel Cell Carcinoma

**DOI:** 10.3390/cancers14205059

**Published:** 2022-10-15

**Authors:** Abbas Rattani, Jeremy Gaskins, Grant McKenzie, Virginia Kate Scharf, Kristy Broman, Maria Pisu, Ashley Holder, Neal Dunlap, David Schwartz, Mehran B. Yusuf

**Affiliations:** 1Department of Radiation Oncology, Tufts University Medical Center, Boston, MA 02111, USA; 2Department of Bioinformatics and Biostatistics, School of Public Health and Information Science, University of Louisville, Louisville, KY 40202, USA; 3Department of Radiation Oncology, Brown Cancer Center, University of Louisville School of Medicine, Louisville, KY 40202, USA; 4School of Medicine, University of Louisville, Louisville, KY 40202, USA; 5Department of Surgery, Division of Surgical Oncology, University of Alabama at Birmingham, Birmingham, AL 35294, USA; 6Department of Medicine, University of Alabama at Birmingham, Birmingham, AL 35294, USA; 7Department of Radiation Oncology, University of Tennessee Health Science Center, Memphis, TN 38163, USA; 8Department of Radiation Oncology, O’Neal Comprehensive Cancer Center, School of Medicine University of Alabama at Birmingham, Birmingham, AL 35294, USA

**Keywords:** Merkel Cell Carcinoma, race, epidemiology

## Abstract

**Simple Summary:**

Merkel Cell Carcinoma is a rare skin cancer most commonly affecting White patients. Less is known for Black patients. We aimed to report presentation, treatment, and quality of registry data by race with a secondary endpoint of overall survival. Findings from this work aim to impact patient and provider awareness to further equitable access to optimal cancer care across the spectrum from diagnosis and treatment to post-therapy surveillance for patients with Merkel Cell Carcinoma.

**Abstract:**

Merkel Cell Carcinoma (MCC) is a rare cancer most commonly affecting White patients; less is known for Black patients. We aim to report presentation, treatment, and quality of registry data by race with a secondary endpoint of overall survival. We conducted a retrospective cohort analysis between 2006–2017 via the National Cancer Database of Black and White MCC patients with and without known staging information. Multivariable logistic, proportional odds logistic, and baseline category logistic regression models were used for our primary endpoint. Multivariable Cox regression was used to interrogate overall survival. Multiple imputation was used to mitigate missing data bias. 34,503 patients with MCC were included (2566 Black patients). Black patients were younger (median age 52 vs. 72, *p* < 0.0001), had higher rates of immunosuppression (28% vs. 14%, *p* = 0.0062), and were more likely to be diagnosed at a higher stage (proportional OR = 1.41, 95% CI 1.25–1.59). No differences were noted by race across receipt of definitive resection (DR), though Black patients did have longer time from diagnosis to DR. Black patients were less likely to receive adjuvant radiation. Black patients were more likely to have missing cancer stage (OR = 1.69, CI 1.52–1.88). Black patients had decreased adjusted risk of mortality (HR 0.73, 0.65–0.81). Given the importance of registry analyses for rare cancers, efforts are needed to ensure complete data coding. Paramount to ensuring equitable access to optimal care for all is the recognition that MCC can occur in Black patients.

## 1. Introduction

Merkel Cell Carcinoma (MCC) is a rare cutaneous malignancy with increasing incidence [1]. While differences by race in baseline presentation, treatment, and outcomes of other skin cancers have been reported [2], such differences for MCC patients have not been completely elucidated. Limited information is available regarding presentation, treatment, or quality of data reporting in national registries for Black patients with MCC. Previous analyses of MCC patients have suggested differences may be present in tumor presentation and overall survival (OS) between Black and White patients [3,4]. Conflicting data exists relating to MCC surgical and radiation treatment by race with some reports suggesting no treatment difference [4], and others reporting differences in terms of time to treatment initiation [5].

Identifying disparities in the diagnosis and treatment of patients with cancer has been federally identified as a critical goal in the United States [6]. This is especially relevant for rare cancer types such as MCC, which less commonly affect non-White patients and subsequently may be lower on a clinician’s differential diagnosis at the time of patient presentation. Further, guideline concordant therapy has been associated with improved outcomes in MCC [7]. Registry analyses are especially important for rare cancers given that level I evidence from randomized trials is often unavailable to guide optimal treatment. However, previous work querying the National Cancer Database (NCDB) for patients with breast, prostate, and lung cancer have shown missing data was more prevalent for Black patients [8].

Thus, we queried the NCDB for MCC patients to determine if baseline presentation, treatment, or data quality differences were present by race—with an examination of potential OS differences as a secondary endpoint. The overarching goal of this inquiry was to improve patient and provider awareness, and encourage equitable access to optimal cancer care across the spectrum of diagnosis and treatment to post-therapy surveillance for patients with MCC.

## 2. Methods

### 2.1. Patient Selection

A retrospective analysis of the NCDB was performed in accordance with the Strengthening the Reporting of Observational Studies in Epidemiology [9] and established race reporting guidelines [10,11] from 2006 to 2017 of all MCC patients with coded White or Black race. Inclusion and exclusion criteria are outlined in Figure 1. Patients with unknown staging were retained in the primary cohort given a primary goal of our analysis was to examine data reporting quality. Importantly, excluding for unknown staging (*n* = 15,000) would have resulted in omission of 41% of White patients and 71% of Black patients (*p* < 0.001).

### 2.2. Outcomes

Baseline demographic and tumor factors were abstracted. Age, sex, race, and additional covariates of insurance, location (e.g., distance to facility), and income (based on patient zip codes of residence) were included to contextualize socioeconomic elements. Tumor characteristics and data quality, like all variables herein (sans OS) were considered categorical. Treatment-timing variables were created using approximate quartiles. Facility volume, defined by computing the number of MCC cases for each facility, was categorized in tertiles; throughout, we refer to the facility that reports data to the NCDB as the treatment facility. OS was defined as time from diagnosis until death from any cause. Survival analysis included patients diagnosed from 2006 to 2016. OS was not available for patients diagnosed in 2017 for the cohort, hence these patients were excluded from survival analyses (but included in all other analyses).

### 2.3. Statistical Analysis

Patient characteristics were compared by race using chi-squared tests. Associations between race, and clinical and treatment patterns were assessed using multivariable modeling (multivariate logistic, proportional odds logistic, and baseline category logistic regressions) while controlling for potential confounders. For all models, confounders were chosen a priori on established clinical relevance. OS was compared by race using Kaplan–Meier curves and the log-rank test, as well as by using multivariable Cox regression to define predictors associated with OS including race—adjusting for demographic factors, tumor characteristics, and treatment received as confounders. Definitive resection (DR) (including Mohs surgery or wide local excision (WLE), excluding excisional biopsy), radiation, and chemotherapy were considered as time-varying predictors in the Cox regression. The proportional hazards assumption was evaluated by considering Schoenfeld residuals; all correlations were less than 0.05 with proportional hazards assumption non-violated. For all multivariate modeling sensitivity was considered by estimating race effect on OS with different sets of confounders to ensure estimate stability. Multiple imputation with 10 imputed datasets was used in all performed analyses to account for missing data. Statistical significance was set at *p <* 0.05. All analyses were performed using R statistical software (v.4.1.2) with the mice and miceadds packages. A sensitivity analysis was performed by repeating all analyses in a second cohort exclusively containing known cancer staging to investigate for any qualitative differences in the conclusions between the two cohorts.

## 3. Results

### 3.1. Patient Demographics

There were 34,503 MCC patients with coded White or Black race were included in the primary cohort. Other race groups were excluded from this analysis given very small numbers and inconsistencies of other reported races—limiting the ability to detect potential differences in a meaningful and clinically relevant manner. Table 1 outlines cohort demographic details along with marginal/unadjusted comparisons between racial groups. A total of 7.4% (*n* = 2556) patients were Black. White patients were proportionally older (≥70 years old, 55%, *p <* 0.0001) and male (61%, *p* < 0.0001) compared to Black patients (≥70 years old, 18%; male 46%). The median age for White patients was 72 years (IQR: 59–81) compared to 52 years (IQR: 37–65) for Black patients. Black patients had substantially higher rates of immunosuppression than White patients (28% vs. 14%, respectively, *p =* 0.0062).

### 3.2. Tumor Characteristics

Significant staging differences were noted between races (*p <* 0.0001). Relative to White patients, the rate of Stage III diagnosis among Black patients was similar, higher for Stages II and IV, and lower for Stage I (37% vs. 49%). Most White patients had tumors < 2 cm (62%) compared to almost two-thirds of Black patients (63%) with tumor sizes greater than 2 cm (*p* < 0.0001). The most common tumor location for White patients was on the head/neck, while the trunk was more common for Black patients (19% vs. 40% of Black patients, *p* < 0.0001). No statistically significant differences were appreciated in presentation of positive nodes.

In the multivariable models (Table 2) adjusted for demographic characteristics, there was a persistent association between Black race and larger tumor size category (adjusted proportional odds ratio [aPOR] 1.99; 95% confidence interval [CI] 1.77–2.24, *p* < 0.0001; tumor size categories from Table 1). An association between tumor site and size was noted in this model, with head and neck primaries being significantly smaller than truncal tumors. However, the association between race and tumor size persisted after adjustment for subsite and the other confounders noted in the table footnotes. Black patients were more likely to present with higher stage than corresponding White patients (aPOR = 1.41, 95% CI 1.25–1.59, *p <* 0.0001), but with no significant difference in the number of positive nodes (*p* = 0.1561). On multivariable analysis, Black patients were more likely to have a truncal (adjusted odds ratio [aOR] 1.87, 1.66–2.10, *p* < 0.0001) or extremity primary site (aOR = 1.24, 1.10–1.39, *p* = 0.0004) compared to the baseline category of head/neck, relative to White patients.

### 3.3. Treatment Patterns

No significant differences by race were noted across receipt of DR (Table 1) by univariate analysis. However, among patients receiving DR, Black patients had significantly longer times from diagnosis to DR than White patients (*p* < 0.0001). Post-resection, Black patients had higher rates of residual tumor on surgical margins (15% vs. 12%, respectively, *p* < 0.0001), though Black patients were not at higher risk of positive surgical margins by adjusted multivariable modeling (Table 2, aOR 1.20, CI 0.93–1.53, *p* = 0.1566). Black patients had lower rates of chemotherapy (CT) receipt (4% vs. 7%, respectively, *p* < 0.0001). White patients had higher rates of radiation (RT) than Black patients (33% vs. 14%, respectively, *p* < 0.0001) and initiating RT within 60 days (28% vs. 22%, respectively; Black patients were delayed in receiving RT over 110 days at a higher rate than Whites, 41% vs. 24%, respectively, *p* < 0.0001).

No difference in receipt of DR by race was found in the cohort by adjusted multivariable analysis (Table 2, *p* = 0.0828). Black patients were more likely to be in a higher time-to-surgery quartile group than White patients (aPOR = 1.22, 1.09–1.37, *p* = 0.0007), adjusting for tumor subsite, stage, size, and other confounders. Regardless of DR, Black patients were less likely to receive primary site radiation even when controlling for tumor stage, tumor size category, facility, and other demographic characteristics (with surgery: aOR for RT 0.51, 0.42–0.61, *p* < 0.0001; without surgery: aOR for RT = 0.54, 0.45–0.64, *p* < 0.0001). Among those receiving DR and RT, the time from diagnosis to RT initiation was longer for Black patients. Receipt of CT was also lower for Black patients who had DR (aOR = 0.50, 031–0.79, *p* = 0.0031) and did not have DR (aOR = 0.63, 0.48–0.85, *p* = 0.0020); no racial difference in CT timing was detected.

### 3.4. Data Quality

Significant differences in reported information by race were appreciated (adjusting for facility characteristics) (Table 2). Black patients were at increased odds of omission of coding for cancer stage (aOR = 1.69, CI 1.52–1.88, *p* < 0.0001) and nodal information (aOR = 2.10, CI 1.84–2.40, *p* < 0.0001). Black patients were more likely to have tumor size reported than White patients (aOR = 0.87, CI 0.79–0.96, *p* = 0.0072).

### 3.5. Overall Survival

OS data was available for patients diagnosed before 2017, representing 90% of the cohort (N = 30,957). Median follow-up time was 42 months (mo) (IQR 20–75); Black patients had longer median follow-up (52 mo vs. 41 mo, *p* < 0.0001). OS was less for White than Black patients across the entire follow-up period (Figure 2, log-rank *p* < 0.0001). The Kaplan–Meier estimated three-year OS rate for White patients was 71.4% (95% CI: 70.9–71.9%) compared to 87.7% (86.3–89.1%) for Black patients. On multivariate analysis, Black patients had a 27% reduced hazard of death relative to White patients (aHR 0.73, 0.65–0.81, *p* < 0.0001). In addition to Black race, other factors associated with improved or poorer survival on multivariable Cox regression modeling are outlined in Table 3. Age was strongly associated with OS in our cohort, with Black patients as noted above significantly younger than White patients. Importantly, race maintained a significant association with OS adjusting for all factors (including age) included in Table 3. Accounting for alternative/reduced sets of confounders in the OS model did not appreciably change the adjusted hazard rate for race.

### 3.6. Sensitivity Analysis

A sensitivity analysis was performed to assess variation between cohorts with (full) and without (restricted) unknown staging information. A substantially smaller proportion of patients were Black in the restricted cohort (4% vs. 7% in full cohort). The restricted cohort was substantially older (especially Black patients), and were more likely to have government insurance, receive radiation therapy, and present with non-truncal primaries (Appendix A). In contrast to previous results, no significant difference was appreciated in stage at diagnosis between Black and White patients in multivariable modeling (Appendix A), but larger tumors were again observed in Black patients. Among the restricted cohort, no difference in DR receipt by race was shown (Black patients were more likely to receive excisional biopsy or non-definitive resection, but the overall factor for surgical type did not meet significance). Black patients continued to demonstrate longer time to DR. Black patients were less likely to receive RT; they had longer times to initiating RT regardless of surgical intervention, and were less likely to receive CT. OS rates in the restricted cohort were lower than in the full cohort (3-yr OS for White 66.3% vs. 78.1% for Black, *p* < 0.0001) (Appendix A), with roughly equivalent racial difference in multivariate modeling (aHR = 0.74, 0.63–0.86, *p* = 0.0001; Appendix A).

## 4. Discussion

This investigation was performed to determine if baseline differences in presentation, treatment, and data reporting quality—with differences in OS as a secondary endpoint—were present by White or Black race in a large national registry of patients with MCC. We observed a younger median age for Black patients, as well as higher rates of immunosuppression. The younger age for Black patients may also explain why a smaller portion of patients would be eligible for government insurance (e.g., Medicare). Nonetheless, a higher rate of Black patients were found to be uninsured relative to White patients. Black patients also had a higher likelihood of presenting with metastatic disease, larger primary tumors, and primary truncal or limb sites. No difference in receipt of DR by race was shown in our cohort, though Black patients were less likely to receive RT or CT. Lengthier time from diagnosis to DR and time to RT initiation was shown for Black patients. Despite this, Black patients in our cohort showed improved OS relative to White patients—acknowledging the limitations of OS as a meaningful clinical endpoint for MCC relative to cancer-specific survival (which was not available in this registry).

The results of this investigation regarding baseline presentation by patient race are concordant with prior studies. An NCDB study by Tripathi et al. found Black MCC patients to present with higher rates of truncal tumors and advanced stage disease. They also found Black patients to present with poorly differentiated, and anaplastic tumors [5]. A Surveillance, Epidemiology, and End Results (SEER) database analysis of MCC patients by Madankumar et al. found MCC patients living below the median household income ($62,872) were likely to be Black, present with tumors of the head and neck, and at a higher stage [3]. In another SEER analysis by Sridharan et al., Black more than White MCC patients presented with larger tumors and were more likely to have distant metastases [4]. As observed by Madankumar et al., Black patients were more likely to present with truncal tumors whereas White MCC patients typically presented with head and neck tumors; no differences were noted in stage by race [3]. Our similar observation of head and neck primaries being significantly smaller than truncal tumors may be due to earlier patient recognition and subsequent diagnosis of such tumors compared to other subsites. In a multi-continent registry analysis by Stang et al., the incidence of MCC of head and neck subsites—more common among older age—was less associated with Black populations as compared to any other group [12]. It has been recognized that tumors appearing on the head/neck tend to have worse survival relative to other subsites [13,14,15]. Tripathi et al. also found time from diagnosis to definitive surgical treatment to be greater for Black than White patients (in agreement with our findings), and longer for Black patients with stages I and II disease [5].

Despite controlling for confounders, data quality and reporting were poorer in the registry dataset for Black patients in terms of cancer stage and presence of nodal metastases. This finding is highly relevant given data missingness as a known potential source of bias—limiting use of large databases to investigate treatment outcomes [8], and potentially modulating of survival results [16]. Improved data quality is vital to ensuring optimal utility of large databases regarding patterns of care and treatment outcomes, which has been emphasized by publications suggesting suboptimal reporting of patient race and underrepresentation of Black patients in prospective clinical trials [17,18,19]. This is especially relevant for rare cancer types including MCC given registry analyses may assist with guiding treatment in the absence of available higher-level evidence from randomized clinical trials.

In contrast to our results, both SEER analyses by Madankumar et al. and Sridharan et al. found Black race to be associated with poorer OS [3,4]. The NCDB is estimated to capture ~70% of all cancer cases diagnosed in the US, in comparison to the SEER which is estimated to cover 28% of the population [20]. Subsequently, our overall cohort and sensitivity cohort (adjusted for documented stage) have a much larger number of Black patients with MCC than in the cohort investigated by Madankumar et al. (32 patients) or Sridharan et al. (59 patients) [3,4]. Black race did not demonstrate significant association in the multivariable analysis performed by Sridharan et al. adjusting for sex, patient age and primary tumor site [4]. Additionally, prior studies have shown OS to be a suboptimal surrogate for cancer-specific survival for patients with MCC [21,22,23,24]. Optimal consideration of survival for patients with MCC should be evaluated in the context of factors such as immune status [14], Merkel cell polyomavirus (MCPyV) status, equivalent definitions of race, and competing causes of mortality. Black patients in our cohort may have had improved survival secondary to a higher distribution of MCPyV-positive tumors (especially in the setting of higher rates of immunosuppression as observed in our cohort) when compared to MCPyV-negative tumors, which have been associated with molecular/mutational profiles consistent with UV-induced carcinogenesis [25]. Unfortunately, this factor is not a recorded variable in the NCDB and thus we were unable to adjust for this in our analyses. For these reasons, OS was not chosen as a primary endpoint for this study. We observed several factors to be associated with improved OS. Lower OS was associated with receipt of RT and chemotherapy, which may reflect substantial selection bias (i.e., patients with poor prognosis may be more likely to receive these treatments, yielding higher adverse treatment effects). In the multivariable Cox modeling (Table 3), we controlled for age, treatment, tumor stage at diagnosis, and the other listed characteristics; thus, the aHR for race represented the relative survival for a Black patient who has the same age, treatment, tumor stage, etc as a White patient. That is, this effect controlled for these characteristics (lower RT/CT, higher stage) that are overrepresented among Black patients.

### Limitations

Limitations of this study merit further discussion. This study is a retrospective analysis of a national cancer registry and confined to its categorization of variables and populations. The age differences observed between White and Black patients should be taken into consideration when evaluating our findings. Data quality and availability from national registries is of critical concern given the large proportion of patients with missing cancer stage and the racial imbalance in the missing data. Patients receiving treatment (s) at facilities other than the one that reports data to NCDB may under- or misreport the full set of interventions a patient receives. Nonetheless, we used multiple imputation and a secondary cohort for sensitivity analysis to mitigate these effects. Additionally, only RT to the primary site was evaluated in this study. We encourage discretion in interpretation. Race is a socially constructed marker and not a biologic or genetic surrogate, hence race presented herein should be viewed from the unique historical context of race in America and the downstream remnants manifesting partially as health inequities [10,11].

While the results presented in this work are intended primarily to be hypothesis generating, we anticipate our findings will improve patient and clinician awareness of MCC as a cutaneous malignancy, which can affect Black patients. We believe our results will motivate clinicians to consider differences in the presentation of MCC across diverse patient groups and consider equitable access to optimal care—especially given the importance of early detection and treatment. Further studies including diverse MCC patient datasets with increased patient, tumor, and treatment granularity (e.g., MCPyV status and cancer-specific endpoints) will be needed to validate the results of this study and provide additional nuance regarding potential differences by race.

## 5. Conclusions

Important differences in baseline presentation, patterns of care, and national registry data reporting quality were demonstrated by race for White and Black patients with MCC in our study cohort. The findings are of high clinical relevance as further support for increased patient and provider awareness that MCC is a cutaneous malignancy also affecting Black patients. Increased awareness is a critical component in the process of ensuring optimal patient access to and utilization of high-quality oncologic care spanning the care spectrum from early/timely diagnosis to guideline concordant post-treatment surveillance for patients with MCC. These results emphasize the need for continued and persistent data quality monitoring efforts for large national registries such as the NCDB. Analyses of such registries regarding patterns of care and clinical outcomes may be of particular value for rare cancers including MCC where level I evidence may be unavailable, with complete data reporting paramount to mitigating missing data bias for these patients.

## Figures and Tables

**Figure 1 cancers-14-05059-f001:**
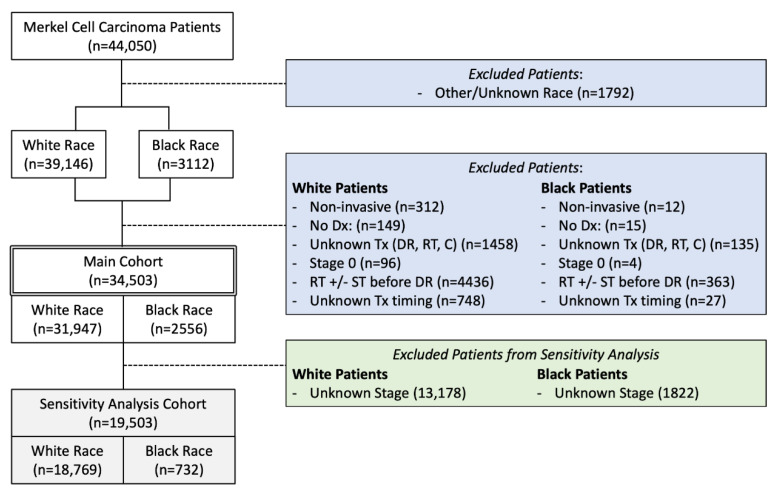
Flow Chart of Study Inclusion Criteria. C = Chemotherapy, DR = Definitive resection, Dx = Diagnostic confirmation, RT = Radiation therapy, ST = Systemic therapy, Tx = Treatment, +/− = with or without.

**Figure 2 cancers-14-05059-f002:**
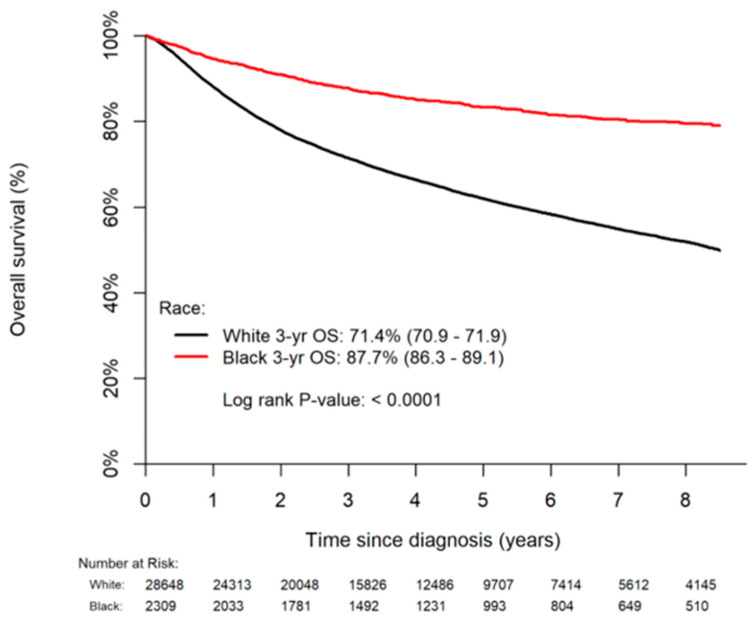
Kaplan–Meier Curve for Overall Survival by Race.

**Table 1 cancers-14-05059-t001:** Cohort characteristics by race.

		Total		White		Black		
		N	Prop	N	Prop	N	Prop	*p*-Value
		34,503		31,947	93%	2556	7%	
**Age**								** *<0.0001* **
	49 or less	5439	16%	4239	13%	1200	47%	
	50–59	4267	12%	3782	12%	485	19%	
	60–69	6729	20%	6313	20%	416	16%	
	70–79	8751	25%	8456	26%	295	12%	
	80+	9317	27%	9157	29%	160	6%	
**Sex**								** *<0.0001* **
	Male	20,528	59%	19,350	61%	1178	46%	
	Female	13,975	41%	12,597	39%	1378	54%	
**Insurance Status**								** *<0.0001* **
	Private Insurance	11,769	35%	10,772.8	34%	1202.5	47%	
	Not Insured	763	2%	588.6	2%	198.9	8%	
	Government	21,427	63%	20,585.6	64%	1154.6	45%	
	Unknown	544						
**Analytic Stage**								** *<0.0001* **
	Stage I	9555	49%	15,752.5	49%	925.9	36%	
	Stage II	3914	20%	6622	21%	792.2	31%	
	Stage III	4602	24%	7227.7	23%	543.3	21%	
	Stage IV	1432	7%	2344.8	7%	294.6	12%	
	Unknown	15,000						
**Definitive Resection**								*0.3833*
	No Definitive Resection	17,450	51%	16,179	51%	1271	50%	
	Definitive Resection	17,053	49%	15,768	49%	1285	50%	
**Definitive Resection (within first 42 days)**								*0.4932*
	No Definitive Resection	23,657	69%	21,810	68%	1847	72%	
	Definitive Resection	10,846	31%	10,137	32%	709	28%	
**Time to Definitive Resection Quartile**								** *<0.0001* **
	Within 21 days	4765	28%	4431	28%	334	26%	
	22–35 days	4330	25%	4066	26%	264	21%	
	36–56 days	4160	24%	3890	25%	270	21%	
	Greater than 57 days	3798	22%	3381	21%	417	32%	
	No Definitive Resection	17,450		16,179		1271		
**Radiation**								** *<0.0001* **
	No Radiation	23,775	69%	21,583	68%	2192	86%	
	Radiation	10,728	31%	10,364	32%	364	14%	
**Time to Radiation Quartile**								** *<0.0001* **
	Within 60 days	2934	27%	2855	28%	79	22%	
	61–80 days	2402	22%	2341	23%	61	17%	
	81–110 days	2710	25%	2636	25%	74	20%	
	Greater than 110 days	2682	25%	2532	24%	150	41%	
	No RT	23,775		21,583		2192		
**Chemotherapy**								** *<0.0001* **
	No Chemo	32,049	93%	29,606	93%	2443	96%	
	Chemotherapy	2454	7%	2341	7%	113	4%	
**Time to Chemo Quartile**								*0.8801*
	Within 30 days	520	21%	495	21%	25	22%	
	31–60 days	701	29%	671	29%	30	27%	
	61–100 days	625	25%	598	26%	27	24%	
	Greater than 100 days	608	25%	577	25%	31	27%	
	No Chemo	32,049		29,606		2443		
**Charlson/Deyo Comorbidity Score**								** *0.0323* **
	0	27,188	79%	25,122	79%	2066	81%	
	1	5101	15%	4757	15%	344	13%	
	2 or more	2214	6%	2068	6%	146	6%	
**Distance to Treatment Center**								** *<0.0001* **
	Less than 10 mi	13,880	45%	13,560.2	42%	1572.7	62%	
	10–25 mi	8487	27%	8860.3	28%	574.1	22%	
	25–50 mi	4210	14%	4575.7	14%	199.9	8%	
	50–100 mi	2657	9%	2916.7	9%	126.4	5%	
	100 mi or more	1849	6%	2034.1	6%	82.9	3%	
	Unknown	3420						
**Zip Code Income Level**								** *<0.0001* **
	$63,000+	11,407	37%	12,120	38%	492.3	19%	
	$48,000–63,000	8470	27%	8901.3	28%	527	21%	
	$38,000–48,000	6904	22%	7096.2	22%	598	23%	
	$0–38,000	4263	14%	3829.5	12%	938.7	37%	
	Unknown	3459						
**Diagnosis Year**								*0.0705*
	2006–2008	6883	20%	6324	20%	559	22%	
	2009–2011	7769	23%	7201	23%	568	22%	
	2012–2014	9185	27%	8510	27%	675	26%	
	2015–2017	10,666	31%	9912	31%	754	29%	
**Tumor Size**								** *<0.0001* **
	No mass found/Microscopic	554	3%	1209.3	4%	32.8	1%	
	Less than 2 cm	11,323	56%	18,606.2	58%	916.7	36%	
	2–5 cm	6135	30%	9132.6	29%	1001.2	39%	
	Greater than 5 cm	2221	11%	2998.9	9%	605.3	24%	
	Size unknown	14,270						
**Resection Margins**								** *<0.0001* **
	No Residual Tumor	26,339	88%	25,401.2	88%	2005.2	84%	
	Residual Tumor	3630	12%	3510.8	12%	373.8	16%	
	No Definitive Resection	3212		3035		177		
	Definitive Resection w/unknown margins	1322						
**Primary Site**								** *<0.0001* **
	Head/Neck	14,812	43%	14,199	44%	613	24%	
	Trunk	7164	21%	6132	19%	1032	40%	
	Limbs	10,879	32%	10,034	31%	845	33%	
	Other/NOS	1648	5%	1582	5%	66	3%	
**Immune Suppression**								** *0.0062* **
	Immunocompetent	5692	89%	27,386.1	86%	1847.8	72%	
	Immunosuppressed	680	11%	4560.9	14%	708.2	28%	
	Unknown	28,077						
**Positive Lymph Nodes**								*0.4507*
	None positive	7285	62%	22,584.3	71%	1806.1	71%	
	1–3 positive	3577	30%	7468.6	23%	626.6	25%	
	4–8 positive	603	5%	1170	4%	77.3	3%	
	9 or more positive	378	3%	724.1	2%	46	2%	
	Unknown	22,660						
**Facility Type**								** *<0.0001* **
	Academic/Research Program	15,206	48%	15,451	48%	1409.1	55%	
	Community Cancer Program	12,759	41%	12,978.6	41%	788	31%	
	Integrated Network Program	3525	11%	3517.4	11%	358.9	14%	
	Facility Type Suppressed (age < 40) ^a^	3013						
**Facility Case Volume**								** *<0.0001* **
	Lowest Third	11,431	33%	10,548	33%	883	35%	
	Middle Third	11,470	33%	10,494	33%	976	38%	
	Largest Third	11,602	34%	10,905	34%	697	27%	

^a^ To ensure patient privacy, NCDB suppressed facility type for all patients with age < 40. For these patients, facility is included in the multiple imputation step as with all other missing data. Variables with unknown categories were imputed using 10 multiple imputation datasets. Cell counts for overall cohort report number of patients with unknown value, but percentages show proportion among those with known stage. The cell counts and percentage values stratified by race are based on the imputed datasets, as this better represents the population distribution for these factors. Any sets that do not add to 100% comes from rounding each percentage to whole numbers. Chemo: Chemotherapy; cm: centimeter; mi: mile (s); NOS: Not otherwise specified; Prop: proportion; RT: Radiation therapy. Bold signifies a new variable as well as *p*-value significance.

**Table 2 cancers-14-05059-t002:** Race Effects from Multivariate Modelling of other Clinical Outcomes.

		Race Effect	Confidence Interval	*p*-Value
**Disease Characteristics**				
Cancer Stage ^A,1^		1.41	1.25–1.59	** *<0.0001* **
Tumor Size ^A,1^		1.99	1.77–2.24	** *<0.0001* **
Number of Positive Nodes ^A,1^		1.14	0.96–1.38	*0.1561*
Subsite ^B,2^				** *<0.0001* **
	Head/Neck	ref		
	Trunk	1.87	1.66–2.10	** *<0.0001* **
	Limbs	1.24	1.10–1.39	** *0.0004* **
	Other/NOS	0.86	0.66–1.13	*0.2950*
**Treatment Characteristics**				
Facility Volume ^A,3^		1.26	1.16–1.38	** *<0.0001* **
Definitive Resection (within 42 days) ^C,4^		0.85	0.77–0.94	** *0.0011* **
Time to Definitive Resection Quartile ^A,4,^*		1.22	1.09–1.37	** *0.0007* **
Positive Surgical Margins (DR pts only) ^C,4^		1.20	0.93–1.53	*0.1566*
Radiation (DR pts only) ^C,4^		0.51	0.42–0.61	** *<0.0001* **
Time to Radiation Quartile (DR pts only) ^A,4,^*		1.80	1.30–2.50	** *0.0004* **
Radiation (no DR pts only) ^C,4^		0.54	0.45–0.64	** *<0.0001* **
Time to Radiation Quartile (no DR pts only) ^A,4,^*		1.30	0.98–1.73	*0.0645*
Chemo (DR only) ^C,5^		0.50	0.31–0.79	** *0.0031* **
Time to Chemo Quartile (DR pts only) ^A,5,^*		2.31	0.97–6.14	*0.0933*
Chemo (no DR pts only) ^C,5^		0.63	0.48–0.85	** *0.0020* **
Time to Chemo Quartile (no DR pts only) ^A,5,^*		1.16	0.77–1.76	*0.4723*
**Data Quality**				
Missing Cancer Stage ^C,6^		1.69	1.52–1.88	** *<0.0001* **
Missing Tumor Size ^C,6^		0.87	0.79–0.96	** *0.0072* **
Missing Nodal Information ^C,6^		2.10	1.84–2.40	** *<0.0001* **

The model implemented depends on the type of outcome variable considered: A. Ordinal logistic regression for ordinal outcomes (race effect is the proportional odds ratio) B. Baseline categorical logistic regression for unordered categorical outcomes (race effect is the odds ratio relative to ref baseline) C. Binary logistic regression for binary outcomes (race effect is odds ratio) * Represents a model only among patients who receive that treatment The set of confounders for each model was chosen a priori based on clinical relevance. The included confounders are the following: 1. Confounders: age, sex, insurance, distance to treatment facility, Charlson/Deyo comorbidity score, diagnosis year, zip code income, subsite, immunosuppression 2. Confounders: age, sex, insurance, distance to treatment facility, Charlson/Deyo comorbidity score, diagnosis year, zip code income, immunosuppression 3. Confounders: age, sex, insurance, distance to treatment facility, Charlson/Deyo comorbidity score, diagnosis year, zip code income, subsite, immunosuppression, stage, tumor size 4. Confounders: age, sex, insurance, distance to treatment facility, Charlson/Deyo comorbidity score, diagnosis year, zip code income, subsite, immunosuppression, stage, tumor size, facility volume, facility type 5. Confounders: age, sex, insurance, distance to treatment facility, Charlson/Deyo comorbidity score, diagnosis year, zip code income, subsite, immunosuppression, stage, tumor size, facility volume, facility type, radiation 6. Confounders: age, sex, insurance, distance to treatment facility, Charlson/Deyo comorbidity score, diagnosis year, zip code income, subsite, immunosuppression, stage, tumor size, facility volume, facility type, radiation, chemotherapy (within 90 day) Chemo: chemotherapy; DR: Definitive resection; pts: patients; NOS: Not otherwise specified; RT: Radiation therapy. Bold signifies a new variable as well as *p*-value significance.

**Table 3 cancers-14-05059-t003:** Multivariate Cox regression modeling for overall survival.

			Adj HR	95% CI		*p*-Values	
**Race**						** *<0.0001* **	
	White		Reference				
	Black		0.73	0.65	0.81		** *<0.0001* **
**Age**						** *<0.0001* **	
	49 or less		Reference				
	50–59		2.93	2.54	3.37		** *<0.0001* **
	60–69		4.42	3.87	5.05		** *<0.0001* **
	70–79		7.16	6.26	8.19		** *<0.0001* **
	80+		14.92	13.05	17.07		** *<0.0001* **
**Sex**						** *<0.0001* **	
	Male		Reference				
	Female		0.78	0.75	0.81		** *<0.0001* **
**Insurance Status**						** *<0.0001* **	
	Private Insurance		Reference				
	Not Insured		1.38	1.15	1.66		** *0.0006* **
	Government		1.35	1.27	1.43		** *<0.0001* **
**Analytic Stage**						** *<0.0001* **	
	Stage I		Reference				
	Stage II		1.19	1.10	1.30		** *<0.0001* **
	Stage III		1.29	1.10	1.51		** *0.0013* **
	Stage IV		1.88	1.62	2.20		** *<0.0001* **
**Procedure (time-varying covariate)**						** *<0.0001* **	
	No Procedure		Reference				
	Excision/Biopsy/Other		0.64	0.59	0.68		** *<0.0001* **
	Mohs		0.56	0.51	0.62		** *<0.0001* **
	Wide Local Excision		0.61	0.57	0.65		** *<0.0001* **
**Radiation (time-varying covariate)**						** *0.0065* **	
	No Radiation		Reference				
	Radiation		1.06	1.02	1.10		** *0.0065* **
**Chemotherapy (time-varying covariate)**						** *<0.0001* **	
	No Chemo		Reference				
	Chemotherapy		1.66	1.55	1.78		** *<0.0001* **
**Charlson/Deyo Comorbidity Score**						** *<0.0001* **	
	0		Reference				
	1		1.25	1.19	1.32		** *<0.0001* **
	2 or more		1.81	1.70	1.93		** *<0.0001* **
**Distance to Treatment Center**						0.6678	
	Less than 10 mi		Reference				
	10–25 mi		1.01	0.96	1.06		0.7172
	25–50 mi		0.96	0.90	1.02		0.2219
	50–100 mi		0.99	0.91	1.07		0.7285
	100 mi or more		0.97	0.89	1.07		0.5535
**Zip Code Income Level**						** *<0.0001* **	
	$63,000+		Reference				
	$48,000–63,000		1.04	0.99	1.10		0.0955
	$38,000–48,000		1.11	1.05	1.16		** *0.0002* **
	$0–38,000		1.15	1.07	1.22		** *<0.0001* **
**Diagnosis Year**							** *<0.0001* **
	2006–2008		Reference			** *<0.0001* **	
	2009–2011		1.03	0.97	1.08		0.3151
	2012–2014		0.98	0.93	1.03		0.4569
	2015–2016		0.86	0.81	0.92		** *<0.0001* **
**Tumor Size**						** *<0.0001* **	
	No mass found/Microscopic		Reference	Reference			
	Less than 2 cm		1.30	1.10	1.53		** *0.0025* **
	2–5 cm		1.37	1.16	1.63		** *0.0003* **
	Greater than 5 cm		1.60	1.33	1.92		** *<0.0001* **
**Resection Margins**						** *<0.0001* **	
	No Residual Tumor		Reference	Reference			
	Residual Tumor		1.35	1.27	1.43		** *<0.0001* **
**Primary Site**						** *<0.0001* **	
	Head/Neck		Reference				
	Trunk		0.84	0.79	0.89		** *<0.0001* **
	Limbs		0.86	0.82	0.90		** *<0.0001* **
	Other/NOS		0.84	0.74	0.96		** *0.0103* **
**Immune Suppression**						** *0.0001* **	
	Immunocompetent		Reference				
	Immunosuppressed		1.15	1.07	1.23		** *0.0001* **
**Positive Lymph Nodes**						** *0.0080* **	
	None positive		Reference				
	1–3 positive		1.11	0.95	1.31		0.1841
	4–8 positive		1.25	1.01	1.54		** *0.0382* **
	9 or more positive		1.53	1.20	1.96		** *<0.0001* **
**Facility Type**						0.1433	
	Academic/Research Program		Reference				
	Community Cancer Program		1.06	1.00	1.12		** *0.0489* **
	Integrated Network Program		1.04	0.97	1.11		0.3267
**Facility Case Volume**						** *<0.0001* **	
	Lowest Third		Reference				
	Middle Third		0.96	0.91	1.01		0.0882
	Largest Third		0.86	0.80	0.92		** *<0.0001* **

Adj HR: Adjusted Hazard Ratio; Chemo: Chemotherapy; CI: Confidence Interval; cm: centimeter; mi: mile (s); NOS: Not otherwise specified. Bold signifies a new variable as well as *p*-value significance.

## Data Availability

All data herein was derived directly from the National Cancer Database (NCDB).

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
