# Peer review of "Patterns of Care and Data Quality in a National Registry of Black and White Patients with Merkel Cell Carcinoma"

_cancers, 2022, doi:10.3390/cancers14205059_

Round 1

Reviewer 1 Report

The manuscript titled “Patterns of Care and Data Quality in a National Registry of Black and White Patients with Merkel Cell Carcinoma” is a retrospective study analyzing whether there is a baseline difference in the presentation, treatment, data reporting quality and survival between Whites or Blacks with Merkel cell carcinoma (MCC). Rattani et al., for the most part show similar results to those published previously. The novelty of this report is that the group shows overall survival is higher in Blacks with Merkel cell carcinoma compared to Whites and that Blacks were less likely to receive radiation and chemotherapy. This study can be used to help inform Blacks that they can get skin cancers such as MCC and hope move patient care to being equitable for all races. Therefore, with edits this manuscript should be ready for publication.   

Suggestion:

-It is very interesting that the authors found a higher over survival rate for Blacks, as no other study of this kind has reported that results. In fact, the authors had similar findings when it came to stage at diagnosis, time to treatment and lack of adequate data reporting in the database. Therefore, it is hard to believe that Blacks, which were less likely to receive radiation or chemotherapy, be at a higher stage at diagnosis, have higher rates of immunosuppression and have a longer diagnosis to definitive resection time would have a higher survival rate.

-In their cox multivariant analysis, did the authors account for the vast difference in age at diagnosis between Whites and Blacks?

-As other groups have shown that over survival in Blacks in reduce compared to Whites in MCC, the authors should explain this difference in findings instead of indicating that over survival is a weakness for this type of study.

-The authors should expand on the methods for their statistic for clarity of the data. For example, page 3 line 98-99 “adjusted for all relevant factors.” A list of what the authors are consider relevant factors would be very helpful here. Page 3 line 104-105 what cofounders are the authors referencing?

One minor suggestion: The authors should place Table 1 before Table 2 in the paper.

Author Response

REVIEWER 1

Comment 1:

It is very interesting that the authors found a higher over survival rate for Blacks, as no other study of this kind has reported that results. In fact, the authors had similar findings when it came to stage at diagnosis, time to treatment and lack of adequate data reporting in the database. Therefore, it is hard to believe that Blacks, which were less likely to receive radiation or chemotherapy, be at a higher stage at diagnosis, have higher rates of immunosuppression and have a longer diagnosis to definitive resection time would have a higher survival rate. In their cox multivariant analysis, did the authors account for the vast difference in age at diagnosis between Whites and Blacks?

Response 1:

This is an important point that the reviewer raises. Yes, as noted in Table 3, the Cox regression model controlled  for age (along with many other confounders).  We agree as expected age is strongly associated with OS in our cohort and this helps explain the magnitude of the (marginal, unadjusted) effect in Figure 2 where Black patients have longer survival.  In the multivariable Cox modeling, we are controlling for age, treatment, tumor stage at diagnosis, and all of the characteristics in Table 3, so the aHR for Race here represented the relative survival for a Black patient who has the same age, treatment, tumor stage, etc as a White patient.  That is, this effect is controlling for these characteristics (lower RT/CT, higher stage) that are overrepresented among Blacks. We have aimed to make this point clearer in the “Overall Survival” subsection of Results and in the Discussion.

Comment 2:

As other groups have shown that over survival in Blacks in reduce compared to Whites in MCC, the authors should explain this difference in findings instead of indicating that over survival is a weakness for this type of study.

Response 2:

Thank you for your thoughtful and insightful comments regarding the benefits of further discussion regarding the OS outcomes in our cohort compared to those in the cohorts queried by Madankumar et al. and Sridharan et al. Both cohorts queried a different registry (the National Cancer Institute’s Surveillance, Epidemiology and End Results or SEER) with much smaller numbers of patients with MCC overall and particularly patients identifying as black (32 patients in Madankumer er al.’s cohort, and 59 patients in Sridharan et al.’s cohort) in comparison to our cohort which includes over 2500 MCC patients identifying as black. Further, when adjusting for other relevant factors including sex, age, and primary tumor anatomic subsite, black race was no longer associated with OS in their MVA (p=.3169) which we have clarified in the text of our manuscript.

Comment 3a:

The authors should expand on the methods for their statistic for clarity of the data. For example, page 3 line 98-99 “adjusted for all relevant factors.” A list of what the authors are consider relevant factors would be very helpful here.

Comment 3b:

Page 3 line 104-105 what cofounders are the authors referencing?

Response 3:

3a: We have reworded this sentence to clarify.  We further note that all confounders included are shown in Table 3, and we have adjusted the heading of the table to clarify that this a multivariate Cox model.

3b: We have fit a number Cox models using different sets of confounders to evaluate the difference in the estimated race effect under these different choices.  As a sample of a few of the models considered: (A) Adjusting only for age: aHR=0.85, 0.77 – 0.95, p=0.003; (B) Adjusting for age and stage only: aHR=0.76, 0.68 – 0.85, p<0.001; (C) Adjusting for confounders associated with treatment facility and treatment received (surgical procedure, RT, CT, distance to treatment, margins, facility type, facility size): aHR=0.71, 0.64 – 0.79, p<0.001; (D) Smaller model with only the most important confounder (race, age, stage, surgery, RT, charlson, facility size): aHR=0.73, 0.66 – 0.82, p<0.001. As all models considered that accounted for at least age and stage produced an aHR in the range of 0.70 to 0.80, we believe that our estimate race effect is not materially impacted by differences in the choice of confounders.

Comment 4:

One minor suggestion: The authors should place Table 1 before Table 2 in the paper.

Response 4:

Thank you for noticing this issue, we have made the revision. 

Reviewer 2 Report

The objective of this study is to compare characteristics and survival of Merkel Cell Carcinoma between black and white patients. The main strength of the study is the great number of participants. Nevertheless, there are several misunderstandings in data results.

The abstract should be adapted to the rule of the journal: The abstract should be a single paragraph and should follow the style of structured abstracts, but WITHOUT HEADINGS.

Table 2 is allocated before table 1.

I most of the studies black raze is a bad prognosis factor. Why do you think the survival is higher is black patients than in white participants in your study?

Black participants were 20 years younger than white participants. Don’t you think that differences in survival are likely due to age instead of race? Could you give any explanation for this great difference in the age of the groups?

Immunosuppression and receiving radiotherapy are also risk factor for aggressive MCC. They are higher in black participants. Moreover, aggressive stages is also higher in black participants. How do you explain that black participants have a higher survival if their MCC they have more frequently aggressive MCC?

Regarding insurance status is also curious that it is higher in black than in white participants.

You obtained many statistical differences likely due to the high number of participants but do you think all of them are relevant?

Why do you use different confounder for each analysis?

Author Response

REVIEWER 2

Comment 1:

The abstract should be adapted to the rule of the journal: The abstract should be a single paragraph and should follow the style of structured abstracts, but WITHOUT HEADINGS.

Response 1:

Thank you for this reminder, we have re-structured the abstract.

Comment 2:

Table 2 is allocated before table 1.

Response 2:

Thank you for noticing this issue, we have made the revision.

Comment 3:

I most of the studies black raze is a bad prognosis factor. Why do you think the survival is higher is black patients than in white participants in your study?

Response 3:Thank you for your thoughtful feedback and perspective regarding further discussion of how our cohort differs from prior work studying race in MCC. We have modified the text of our revised manuscript to highlight our cohort contains a much larger sample size of overall patients with MCC as well as those identifying as black(>2500)  in comparison to those queried by Madankumer et al (32) or Sridharan et al (49). We also highlighted differences in the NCDB vs SEER.

Black patients in our cohort may have improved survival secondary to an imbalanced distribution of MCPyV + tumors in comparison to MCPyV negative tumors which have been associated with molecular/mutational profiles consistent with UV-induced carcinogenesis potentially less common by melanin. Unfortunately, this factor is not available in the dataset and we were unable to adjust for this in our analyses.  We have added this observation to the Discussion section.

Given the limitations of survival analysis using the NCDB which only captures details regarding the initial course of treatment, and lacks cancer specific endpoints including information regarding recurrence, and disease specific survival, we have emphasized our findings regarding survival are hypothesis generating but we are unable to suggest or demonstrate causation underlying this difference in OS appreciated by race.

Comment 4:

Black participants were 20 years younger than white participants. Don’t you think that differences in survival are likely due to age instead of race? Could you give any explanation for this great difference in the age of the groups?

Response 4:

Thank you for your insightful comments regarding the difference in age noted by race in our cohort. We agree age is strongly associated with OS in our cohort and that this helps explain the (marginal/unconditional) effect in Figure 2 where Black patients have longer survival.  In the multivariable Cox modeling, we are controlling for age, treatment, tumor stage at diagnosis, and all of the characteristics in Table 3, so the aHR for Race here represent the relative survival for a Black patient who has the same age, treatment, tumor stage, etc as a White patient.  That is, this effect is controlling for these characteristics (lower RT/CT, higher stage) that are overrepresented among Blacks. We have added this clarification to the Discussion section of our paper as well.

With regards to the difference in age, we note black patients in our cohort have higher rates of immunosuppression which in itself could contribute to earlier onset/age of diagnosis. We speculate black patients in our cohort may also have higher rates of MCPyV+ though this factor was not available in the dataset and we are unable to analyze this. These reflections have also been incorporated in the Discussion section.

Comment 5:

Immunosuppression and receiving radiotherapy are also risk factor for aggressive MCC. They are higher in black participants. Moreover, aggressive stages is also higher in black participants. How do you explain that black participants have a higher survival if their MCC they have more frequently aggressive MCC?

Response 5:

Thank you for the above comments. Black patients in our cohort were less likely to receive radiation therapy, and when receiving radiation were at higher risk of delayed time to initiation. Black patients did have higher rates of IS in our cohort, but we do not have granularity in the dataset to adjust for a quantitative measure of severity of IS by examining ANC or CD4 count.

Importantly, we also lack MCPyV status in our cohort. All these factors may influence disease specific survival, which unfortunately is not available as an endpoint to analyze. We emphasize caution with examining OS given OS and disease specific survival may have poor concordance for older patients or patients with early-stage disease, and with the NCDB unable to account for courses of therapy beyond the initial treatment course. These reflections have also been stated in the Discussion section of our revised draft.

Comment 6:

Regarding insurance status is also curious that it is higher in black than in white participants.

Response 6:

Thank you for your valuable feedback. Black patients due to significantly younger median age were less likely to be eligible for government insurance (Medicare), which likely accounts for the higher rate of private insurance. Interestingly, black patients did have higher rates of being uninsured in our cohort relative to white patients. This point was clarified early in our Discussion section.

Comment 7:

You obtained many statistical differences likely due to the high number of participants but do you think all of them are relevant?

Response 7:

For all effects in Tables 2 and 3, we have relevant effect sizes for the race effects (in terms of adjusted odds ratios, adjusted proportional odds ratio, or adjusted hazard ratios), so that the reader can judge the magnitude of these effects to determine the scientific significance.  We have limited our discussion in the manuscript of those effects that we believe are of a magnitude to be both statistically significant and clinically relevant.

Comment 8:

Why do you use different confounder for each analysis?

Response 8:

In Table 2, it is important that we choose a set of confounders for each outcome variable that is consistent with a plausible causal interaction.  For instance, when the outcome variable is definitive resection, we want to include tumor size as the clinical decision to resect or not will depend on this tumor characteristic.  Conversely, when we model tumor size, we would not want to control for resection, as it is implausible that the later decision to resect impacts the tumor size.  While we are not necessarily advocating a causal interpretation of these model results, it is important that we limit ourselves to predictors that are consistent with plausible causality.  The choices in this table are determined by our choice the disease characteristics should depend on demographic characteristics (but treatment characteristics) and treatment decisions are based on both demographics and disease characteristics. Further, radiation will depend on whether or not the patient is resected and chemotherapy will depend on both RT and DR. We are happy to include this reflection in the Methods section should the reviewer/editor believe it be of value to the reader.

Round 2

Reviewer 2 Report

Thank you for your reply

The mansucript has been improve. You should include in the limitations that black participants were 20 years younger than white participants. It is an important bias that could be greatly impact on survival

Author Response

We thank the reviewer for this suggestion, and we have made this change to the Limitations section in our manuscript.